# The Importance of CXCL1 in Physiology and Noncancerous Diseases of Bone, Bone Marrow, Muscle and the Nervous System

**DOI:** 10.3390/ijms23084205

**Published:** 2022-04-11

**Authors:** Jan Korbecki, Magdalena Gąssowska-Dobrowolska, Jerzy Wójcik, Iwona Szatkowska, Katarzyna Barczak, Mikołaj Chlubek, Irena Baranowska-Bosiacka

**Affiliations:** 1Department of Biochemistry and Medical Chemistry, Pomeranian Medical University, Powstańców Wlkp. 72 Av., 70-111 Szczecin, Poland; jan.korbecki@onet.eu (J.K.); mikolaj.chlubek@gmail.com (M.C.); 2Department of Ruminants Science, Faculty of Biotechnology and Animal Husbandry, West Pomeranian University of Technology, Klemensa Janickiego 29 St., 71-270 Szczecin, Poland; jerzy.wojcik@zut.edu.pl (J.W.); iwona.szatkowska@zut.edu.pl (I.S.); 3Department of Cellular Signalling, Mossakowski Medical Research Institute, Polish Academy of Sciences, Pawińskiego 5, 02-106 Warsaw, Poland; magy80@gmail.com; 4Department of Conservative Dentistry and Endodontics, Pomeranian Medical University, Powstańców Wlkp. 72 Av., 70-111 Szczecin, Poland; katarzyna.barczak@pum.edu.pl

**Keywords:** CXCL1, CXCR2, brain, chemokine, cytokine, neutrophil, CINC-1, KC, Gro-α

## Abstract

This review describes the role of CXCL1, a chemokine crucial in inflammation as a chemoattractant for neutrophils, in physiology and in selected major non-cancer diseases. Due to the vast amount of available information, we focus on the role CXCL1 plays in the physiology of bones, bone marrow, muscle and the nervous system. For this reason, we describe its effects on hematopoietic stem cells, myoblasts, oligodendrocyte progenitors and osteoclast precursors. We also present the involvement of CXCL1 in diseases of selected tissues and organs including Alzheimer’s disease, epilepsy, herpes simplex virus type 1 (HSV-1) encephalitis, ischemic stroke, major depression, multiple sclerosis, neuromyelitis optica, neuropathic pain, osteoporosis, prion diseases, rheumatoid arthritis, tick-borne encephalitis (TBE), traumatic spinal cord injury and West Nile fever.

## 1. Introduction

Chemokines are chemotactic cytokines [1], whose most important function is the chemoattraction of immune cells. These days we have a good understanding of the role each of the almost 50 chemokines identified in humans plays in the immune system and in inflammatory responses. They are crucial for the normal functioning of various organs as well as correct prenatal development, while being involved in the pathogenesis of various diseases, particularly cancer [2]. Chemokines are classified into the following subfamilies according to their characteristic conserved N-terminal cysteine motif:CX3C chemokines (1 representative in humans),CXC chemokines (17 in humans),CC chemokines (26 in humans),XC chemokines (2 in humans).

CXC motif chemokine ligand 1 (CXCL1), a CXC chemokine [1], is also known as growth-regulated (or -related) oncogene-α (Gro-α) [3] and melanoma growth-stimulatory activity (MGSA) [4]. It is one of seven chemokines that activate the CXC motif chemokine receptor 2 (CXCR2) [1] and one of the most studied CXC chemokines. There is a lack of work summarizing the role of this chemokine in the physiology and pathology of non-cancer diseases. Due to the considerable body of knowledge, in this review we focus on bone, muscle and the nervous system.

## 2. Commentary on the Research Methodology

### 2.1. Search and Selection of Articles

This review is based on articles available in PubMed (https://pubmed.ncbi.nlm.nih.gov; accessed on 30 December 2021). The search phrases included: 

(CXCL1 or CINC-1 or MIP-2 or MIP2 or (KC chemokine) or MGSA or gro-a or gro-alpha) and (bone or muscle or brain or astrocyte or microglia or neuron) not review.

The results included all articles with the title, abstract or keywords mentioning selected organs and CXCL1 or paralogs of this chemokine in rodents. Review papers were not searched. Over 1700 articles were searched and preselected based on the title of the paper. Based on the literature searched, subsections were written about the physiological role of CXCL1 in given tissues and organs. After selecting the most exhaustive 100 articles, 15 diseases were selected in which CXCL1, or its mouse or rat paralog, plays an important role. Then to write each of the subsections on the role of CXCL1 in diseases, articles were searched by the phrase: 

“the name of the disease” and (CXCL1 or MIP-2 or MIP2 or (KC chemokine) or MGSA or gro-a or gro-alpha) and not review.

Next, a selection of search articles was made by title. The papers were then read and searched for new and interesing literature references.

In addition, each subsection contains a brief introduction about the disease being discussed. To write these introductions, PubMed (https://pubmed.ncbi.nlm.nih.gov; accessed on 30 December 2021) was used and the phrase: 

“the name of the disease” and review.

The search was performed with the “best match” function.

The selected feature guaranteed that the first 50 results would include a review that discussed all aspects of the disease being searched for. Once one to two papers were selected, they were read and a brief introduction to the subsections was written based on the papers.

### 2.2. The Lack of In Vivo Models for CXCL1 Functions

In this review, we describe the role of CXCL1 in the physiology of selected organs and in selected diseases. A major challenge in writing this review was the lack of in vivo animal experiments in which human CXCL1 was studied. This is associated with the fact that the common ancestor of humans, mice and rats had far fewer CXC chemokine genes than today’s mammals [5,6]. Over the course of evolution, the duplication of genes for CXC chemokines, which are ligands for CXCR2, has resulted in the formation of 7 CXC chemokines that at low concentrations are ligands for CXCR2–*CXCL1*, *CXCL2*, *CXCL3*, *CXCL5*, *CXCL6*, *CXCL7* and *CXCL8*, all of which form a gene cluster [5,6]. CXCL6 and CXCL8/interleukin-8 (IL-8) are also ligands for the CXC motif chemokine receptor 1 (CXCR1) [7]. CXCL1, CXCL2, CXCL3, CXCL5 and CXCL7 do not differ in their biological properties between humans and rodents. The most significant differences between these chemokines are associated with their expression in various physiological and pathological states. The expression of a given CXC chemokine as a ligand for CXCR2 is cell-type dependent, something that has been the result of evolution [8]. For this reason, it is impossible to determine which of the ligands for CXCR2 is involved in a given disease both in humans and experimental animals (mice or rats) [5], or whether the same chemokine is involved in a given disease simultaneously in both humans and rodents.

The genes and proteins of chemokines that are ligands for CXCR2 in the mouse and rat are not described using the CXC [number] standard used for chemokines found in humans. The counterparts of CXCL1 in rodents are cytokine-induced neutrophil chemoattractant-1 (CINC-1) in rats [9,10], and keratinocyte-derived chemokine (KC) in mice [11,12]. Both chemokines, CINC-1 and KC, are not ligands for CXCR1 at low concentrations [13], which makes them similar to human CXCL1. However, mouse KC is located in the part of the chromosome assigned for human chemokines CXCL3, CXCL4, CXCL5 and CXCL7, and given the order of the chemokines in this gene cluster it does not correspond to human CXCL1 [6].

Due to these discrepancies, we adopted the following principles when writing this review. First, to show whether CXCL1 has a role in a given process in humans, we showed elevated CXCL1 expression in patients with a given disease or at the site of given physiological processes. Then, we described the mechanism of CXCL1 action, including in vivo studies on experimental animals with elevated expression of chemokines that are ligands for CXCR2, namely, KC or macrophage inflammatory protein-2 (MIP-2) in the mouse and CINC-1 in the rat. It can be assumed that if KC, MIP-2, CINC-1 and CXCL1 appear in different species in the same disease, then they play the same role in disease mechanisms. Importantly, in the present study we focus only on human CXCL1, although it is possible that other human ligands for CXCR2 also have the same function in the diseases we discuss.

In some studies on mice, KC has a significant function in a given disease, even though in humans CXCL1 does not. In such a case, we describe the mechanism of action of CXCR2 ligands in a given disease and then provide possible CXC chemokines that play some role in humans, most frequently CXCL8/IL-8. The large majority of literature does not examine all ligands for CXCR2 and for this reason it is possible that other CXC chemokines, not just CXCL8/IL-8, may also be involved in the mechanisms of a given disease.

## 3. CXCL1 Action at the Single Cell Level

CXCL1 is a chemoattractant cytokine. Its most important property is to cause chemotaxis of immune cells, mainly neutrophils [14,15,16], and to a lesser extent CD14^+^ monocytes [17] and basophils [18], but not T cells [19]. It also reduces neutrophil apoptosis and thus increases the accumulation of these cells at sites of inflammatory reactions [20]. CXCL1 also causes the chemotaxis of endothelial cells and thus participates in angiogenesis [21,22].

The main receptor for CXCL1 is CXCR2, activated at concentrations of just a few nanomoles of CXCL1 [23,24,25]. At concentrations in the order of 100 nM, CXCL1 can also activate CXCR1 [23,24,25]. CXCL1 can also bind to atypical chemokine receptor 1 (ACKR1)/Duffy antigen receptor for chemokines (DARC) [26] and thus protect against an excessive inflammatory response [27,28].

Any inflammatory response is associated with an increase in the level of pro-inflammatory cytokines which induce an increase in CXCL1 expression. Interleukin-1β (IL-1β) and tumor necrosis factor-α (TNF-α) increase CXCL1 expression at the transcriptional level [29,30]. Interleukin-17 (IL-17) increases CXCL1 mRNA stability [31,32,33]. Then, the cytokine-induced increase in CXCL1 expression leads to the infiltration of neutrophils into the sites of inflammatory responses, which contributes to the fight against pathogens or agents that triggered the inflammatory responses [14,15,16].

CXCL1 may also increase the proliferation of certain cells. CXCL1 was often referred to as an autostimulatory melanoma mitogen, and hence one of its first names, melanoma growth-stimulatory activity (MGSA) [4]. The CXCL1 ability to increase proliferation applies mainly to cancer cells [4,34,35,36,37], and also, for example, to oligodendrocyte precursors [38,39].

CXCL1 is important in cell death. Cell apoptosis is associated with an increase in CXCL1 expression as a result of Fas/CD95—an effect dependent on NF-κB activation [40]. CXCL1 is a chemotactic factor for neutrophils [14,15,16], i.e., it causes the chemotaxis of neutrophils to sites of apoptosis where they remove cell debris. [40]. In this way, CXCL1 constitutes a so-called ‘find-me’ signal. Also, the release of IL-1α from cells during necrosis leads to an increase in CXCL1 expression in mesothelial cells and to the recruitment of neutrophils involved in the clearance of cell debris and necrosis factors [41].

CXCL1 is also important in cellular senescence—inhibition of the cell cycle in response to adverse stressful factors [42]. In this process, NF-κB activation results in changes in the cellular secretion of various factors [43]. Such a cell exhibits the so-called senescence-associated secretory phenotype (SASP), important in the recognition of the cell by the immune system cells, in particular natural killer (NK) cells [43] and CD4^+^ T cells [44]. The CXCL1→CXCR2 axis also plays an important role in senescence. Exposure of cells to adverse factors results in the activation of p53 [45,46], which attaches to the promoter of the *CXCR2* gene, thus increasing the expression of the receptor for CXCL1. Also during senescence, NF-κB activation increases the expression of CXCL1 and other CXCR2 ligands [45,47,48]. The activation of the CXCL1→CXCR2 axis reinforces the growth arrest of cells during senescence [45], a p53-dependent process protecting against cancer development [48]. Significantly, frequent mutations in the tumor protein p53 (*TP53*) gene which encodes p53 lead to changes in the properties of CXCR2. In cells with *TP53* mutations, this receptor does not inhibit but rather increases proliferation, which leads to tumorigenesis [45]. Finally, CXCL1 alone can also cause senescence, which has important implications in malignant tumors. CXCL1 secreted by cancer cells [49,50] causes p53-dependent senescence of fibroblasts which then begin to express SASP, which in a tumor, begins to promote tumor growth.

Knowledge of the action of CXCL1 at the cellular level provides a basis for understanding its action at the level of organs and the entire body. The following sections discuss the role of CXCL1 in the physiology of selected organs and its role in selected non-cancer diseases. Due to the vast body of knowledge in this study, we focus on bone, bone marrow, muscle and nervous tissue.

## 4. Cartilage and Bone Tissue

### 4.1. Bone, Fracture Healing, Osteoporosis

CXCL1 is involved in the physiology and pathology of bone tissue. Most data are based on studies of KC, a mouse paralog for human CXCL1, and for this reason require confirmation in research on humans.

The expression of ligands for CXCR2, such as KC and MIP-2, has been shown to be elevated in murine osteocytes under shear stress [51,52] and by parathormone (PTH) and parathyroid hormone-related protein (PTHrP) [53]. This increase in KC expression causes osteoclast precursors to migrate [51], and the subsequent activation of CXCR2 on these cells enhances osteoclast maturation [54]. This is followed by either bone remodeling or bone resorption under the influence of factors that stimulate KC expression. Human osteoclast precursors exhibit CXCR2 expression [55] but it is reduced during differentiation of these cells into osteoclasts. It appears that CXCL1 may have the same properties in bone as KC, and so may participate in bone modelling in humans, although this should be confirmed by further studies.

Due to the induction of osteoclast maturation by CXCR2 ligands, CXCL1 levels are positively correlated with osteoporosis in humans [56]. Also, bone marrow adipocytes produce ligands for CXCR2 [54], which leads to the weakened bone structure in mice with advanced age or obesity [54]. If a similar mechanism occurs in humans, then this could account for the frequent bone fractures in older people or those with obesity. CXCL1 is also important in fractures—a condition associated with an increase in KC expression in mice [57]. This chemokine is indirectly important in fracture healing, via neutrophils recruited by this chemokine [57].

The expression of ligands for CXCR2 by bone marrow adipocytes may support the formation of cancer bone metastasis in elderly or obese people [54]. Also, the increased expression of ligands for CXCR2 in osteocytes under shear stress and PTHrP may support bone metastasis of some cancers, including breast cancer [52,58]. Not less important is CXCL1 production by cancer cells in bone metastasis [59], as CXCL1 stimulates cancer cell proliferation [35,37,60], as well as participating in bone remodeling during bone metastasis formation [54,59]. If a tumor cell from the blood stops in bone tissue, it causes bone remodeling by secreting CXCL1 and hence bone metastasis.

### 4.2. Bone Marrow

Ligands for CXCR2, including CXCL1, are important in the self-renewal capacity of hematopoietic stem cells [61]. Human CD34^+^CD38^-^ express CXCL1 as well as other ligands for CXCR2 such as CXCL2, CXCL6 and CXCL8/IL-8—chemokines crucial for hematopoietic stem cell maintenance.

CXCL1 is also significant in the regulation of whole body immunity via neutrophil egress from the bone marrow [62,63]. Two axes are responsible for the regulation of neutrophil release from the bone marrow. CXCL12/SDF-1→CXCR4 is responsible for the retention of neutrophils and homing of senescent neutrophils to the bone marrow [62], while CXCR2 ligands are responsible for neutrophil egress from the bone marrow [62,63]. Also, pro-inflammatory factors in the blood, such as LPS, increase CXCL1 expression in endothelial cells in the bone marrow—an effect dependent on β-adrenergic signaling [63] The release of neutrophils under the influence of pro-inflammatory factors in the blood is important in the fight against pathogens. Acute inflammation increases the levels of pro-inflammatory cytokines in the blood leading to the mobilization of neutrophils and subsequent accumulation of these cells at sites of intense inflammatory responses.

Chronic inflammation is associated with elevated levels of ligands for CXCR2, including CXCL1, which cause the expansion of monocytic myeloid-derived suppressor cell (MDSC) in the bone marrow as shown in mice [64]. This is associated with CXCR2 activation on granulocyte and macrophage progenitor cells (GMPs) [65], which reduces the expression of Sin3-associated 18 kDa polypeptide (SAP18). This, in turn, results in the activation of extracellular signal-regulated kinase (ERK) mitogen-activated protein kinase (MAPK) and signal transducer and activator of transcription 3 (STAT3), which increases granulocyte monocyte progenitor (GMP) differentiation into macrophages and dendritic cell progenitor cells (MDP) [65]. Subsequently, in the bone marrow, MDP differentiate into monocytic MDSC, resulting in an increase in the number of these cells. This effect is important in diseases with chronic inflammation. Expansion of monocytic MDSC in the bone marrow results in an increase in the number of these cells in the blood, which leads to an overall weakening of the immune system.

### 4.3. Rheumatoid Arthritis

Rheumatoid arthritis, estimated to affect less than 1% of the human population, is an autoimmune disease that is characterized by chronic inflammation which results in the destruction of joints [66]. One component of the pathophysiology of rheumatoid arthritis is an increase in CXCL1 expression in rheumatoid arthritis patients in the blood [67] and synovial fluid [68,69]. At the same time, CXCL1 expression in synovial fluid is higher in patients with rheumatoid arthritis than in those with osteoarthritis [67,69,70].

CXCL1 in the synovial fluid comes from fibroblast-like synoviocytes (FLS), chondrocytes and neutrophils (Figure 1). In particular, increased CXCL1 expression occurs in the lining layer [71]. FLS increases the expression of CXCL1 under the influence of pro-inflammatory cytokines such as TNF-α and IL-1β [68,72], whose expression is also increased in rheumatoid arthritis patients [73,74]. That means that chronic inflammation in joints increases the expression of TNF-α and IL-1β, which increases the expression of CXCL1. The synovial fluid in patients with rheumatoid arthritis also show increased levels of IL-17 [75], a cytokine that increases CXCL1 expression, particularly in FLS [76]. In FLS, the expression of CXCL1 is also increased by resistin, an adipokine produced by macrophages located in the synovium in patients with rheumatoid arthritis [77].

CXCL1 participates in rheumatoid arthritis by acting on various cells in the joints. It causes hypertrophy of chondrocytes [78] resulting in an elevated expression of MMP-13, an enzyme that degrades collagen and aggrecan. This results in degradation of ECM in articular cartilage followed by apoptosis of chondrocytes and degradation of cartilage in the joints.

CXCL1 also acts on FLS. Although it does not cause the proliferation of FLS [79], it does reduce collagen production in these cells, which interferes with the normal function of these cells in the joints. CXCL1 also increases the production of IL-6 in FLS [69], one of the factors causing an increase in IL-6 in synovial fluid in patients with rheumatoid arthritis; such a response does not occur in healthy individuals [69,72]. IL-6 is a cytokine that is involved in rheumatoid arthritis by causing bone resorption and by participating in inflammatory reactions [80].

CXCL1 causes an ingress of neutrophils into the joints [81,82], a process that also appears to require LTB_4_ [82]. CXCL1 has also been shown to act on neutrophils in the joints by increasing the production and secretion of LTB_4_ in these cells [82]. This bioactive lipid causes an ingress of leukocytes into joints, where they act destructively on joint tissue and thus contribute to the pathogenesis of rheumatoid arthritis. Neutrophils also produce MMP-8 and MMP-9 which degrade collagen [83]. Neutrophils also produce ROS, which have a destructive effect on joint tissue, and various proteases such as elastase, cathepsin G and proteinase-3, which are involved in joint tissue destruction and inflammatory reactions.

CXCL1 can also increase osteoclast activity, which leads to bone erosion [54,55,56]. However, the importance of CXCL1 in the destruction of bone tissue in the joints of patients with rheumatoid arthritis is yet to be thoroughly investigated.

## 5. Muscles

CXCL1 may play an important physiological role, particularly in muscle function. However, due to the lack of an appropriate research model, these are assumptions drawn from a mouse model for changes in KC chemokine expression. Therefore, this physiological aspect requires further studies on humans.

### 5.1. Muscle Physiology

Exercise is associated with an increase in the expression of IL-6 and CXC chemokines that are ligands for CXCR2, such as KC and lipopolysaccharide-induced CXC chemokine (LIX) in the muscle and blood of mice [84,85,86,87,88]. Significantly, the increase in KC expression in the muscle is independent of IL-6 [88]. IL-6 from the muscle travels via the blood to the liver, where the expression of KC increases, which then is responsible for the increase in blood KC levels [86]. KC also acts in an autocrine manner on muscle via CXCR2, which induces an increase in muscle insulin responsiveness, specifically an increase in glucose transporter 4 (GLUT4) recycling [84]. However, the same authors in a later study question the effect of KC and LIX on GLUT4 recycling in muscle [85]. KC also increases fatty acid oxidation [87] and muscle angiogenesis (Figure 2) [87].

KC is also considered a myokine as it causes proliferation, self-renewal of satellite cells and myogenesis from satellite cells—stem cells present in muscles that participate in regeneration [89]. KC is also a chemotactic factor for myoblasts and causes myogenic differentiation of these cells [85]. As a consequence of the action of KC, there is an expansion of the muscle and an increase in muscle efficiency. Also of note is the exercise-induced increase in blood levels of KC in mice, and most likely also CXCL1 in humans [86]. KC and CXCL1 cause the mobilization of neutrophils from the bone marrow, whose function is to destroy pathogens [62]. It can be speculated that exercise in the described mechanism may enhance immunity.

The expression of ligands for CXCR2 is also subject to upregulation in muscle regeneration, as shown by studies in cattle [90]. Their role in muscle regeneration is additionally indicated by the fact that their expression is tightly regulated by myostatin [90]. Further research in this area is required to determine the exact mechanism of muscle regeneration.

### 5.2. Muscle, CXCL1 and Obesity

CXCL1 participates in muscle disease mechanisms. Saturated fatty acids, particularly palmitate, cause myotube loss [91] which is associated with a decrease in the expression of certain myokines. At the same time, palmitate also increases the expressions of CXCL1 in human muscle and KC in mouse muscle [89]. In mice, KC stimulates proliferation and self-renewal of satellite cells [89] and thus it counteracts the negative effects of palmitate on muscle. A similar mechanism may occur in humans—palmitate may increase the expression of CXCL1 in muscle which then inhibits the adverse effect of this acid. This process is of importance as ~60% of the North American and European populations are overweight [92].

### 5.3. Tumor-Induced Muscle Wasting

CXCL1 may also participate in tumor-induced muscle atrophy, one of the components of cancer cachexia [93,94]. Although this review does not focus on cancer, this section shows the effect that chronic inflammation has on muscle, as in advanced cancer. Patients with breast cancer [95], esophageal squamous cell carcinoma [96], ovarian cancer [97] and renal cell carcinoma [98] have elevated levels of CXCL1 in the blood. In addition, studies in mouse models have shown that factors from tumorigenesis increase KC expression in muscle [93]. KC, produced in muscle as well as secreted from a tumor, impairs myoblast differentiation, leading to muscle atrophy. This effect is also enhanced by other factors from the tumor such as insulin like growth factor binding protein 3 (IGFBP3) and CC motif chemokine ligand 2 (CCL2) [93]. Another mechanism by which KC causes tumor-induced muscle atrophy is the infiltration of skeletal muscle by immune cells, including neutrophils and macrophages [93]. These cells suppress myogenic differentiation, leading to tumor-induced muscle atrophy.

## 6. The Nervous System

### 6.1. Prenatal Development of the Brain

CXCL1 may be important in prenatal brain development. However, the significance of CXCL1 in prenatal development in humans has been poorly studied due to obvious bioethical problems with such research. The importance of ligands for CXCR2 in this aspect has been much better studied in animals.

Experiments on rats show that ligands for CXCR2 are important in axon morphogenesis, a process regulated by hepatocyte growth factor (HGF). For example, this factor in rats increases the expression of cytokine-induced neutrophil chemoattractant-3 (CINC-3), a CXC chemokine important in axon outgrowth and axon branching [99]. Ligands for CXCR2 are known to reduce axon outgrowth of dorsal root ganglia neurons in adult rats [100], thereby reducing peripheral nervous system regeneration. However, the significant differences in ligands for CXCR2 between humans and rats make it impossible to determine which human chemokine is responsible for the aforementioned properties [5].

CXCL1 may be important in the functions of oligodendrocyte progenitors. For example, research on rats has shown that CINC-1 increases the proliferation of these cells, exhibiting synergy with platelet-derived growth factor (PDGF) [38,101]. In humans, this role is played by CXCL1 [102]. Expression of this chemokine occurs in the cortical ventricular/subventricular zones, the sites of proliferation of oligodendrocyte progenitors. CXCL1 can inhibit the migration of oligodendrocyte progenitors, as shown by research on the effects of CINC-1 in the rat brain and spinal cord [101,103], indicating that this chemokine regulates the location of oligodendrocytes.

### 6.2. Neurogenesis, Hippocampus and Neural Stem Cells

CXCR2 ligands are important in the hippocampus, particularly for neurogenesis, as demonstrated by experiments in mice and rats. In mice, KC increases hippocampal neurogenesis [104]. The expression of this chemokine occurs in the subgranular zone of the dentate gyrus in the hippocampus [105]. KC induces neural stem cell proliferation but inhibits the differentiation of these cells into astrocytes [106]. At the same time, studies in rats have shown that CINC-1 is subject to expression in damaged parts of the brain, particularly in the damaged striatum, where this chemokine causes the recruitment of progenitor cells from subventricular zone [107], that is, it participates in the regeneration of neural tissue in the brain. Nevertheless, the cited facts are yet to be confirmed in humans.

CXC chemokines can also interfere with hippocampal neurogenesis in some models. In particular, during neuroinflammation [105]. KC in a mouse model causes senescence of hippocampal neuronal progenitor cells. This reduces neurogenesis in this brain structure. These results were confirmed on human hippocampal neuronal progenitor cells and CXCL1 [105]. The observed effect was sex-dependent, as female sex hormones counteract the increase in KC expression by pro-inflammatory factors in the hippocampus [105].

### 6.3. Addiction and Reward System

CXCL1 may also play a role in the mechanisms of addiction. In mice, administration of cocaine increases KC expression in the prefrontal cortex, in a process dependent on the dopamine D_1_ receptor [108]. KC, through the activation of CXCR2, is an important part of reward system activation in the brain when exposed to cocaine. To date, there has been no such research on humans and for this reason it is not known whether CXCL1 or any other ligands for CXCR, is significant in the action of cocaine in the human brain. There are no studies regarding the effect of CXCL1 on the reward system during daily activities [108].

### 6.4. Alzheimer’s Disease

Alzheimer’s disease is a neurodegenerative disease that causes dementia [109]. It is estimated that tens of millions of people suffer from this disease worldwide, mostly in advanced age. An important pathomechanism in Alzheimer’s disease is the accumulation of amyloid β (Aβ) in amyloid plaques and tau protein in neurofibrillary tangles, which cause neurodegeneration of the brain. Another component of Alzheimer’s disease is neuroinflammation, including increased CXCL1 expression and action.

In Alzheimer’s disease patients, CXCL1 levels are elevated in cerebrospinal fluid [106,110] and the brain [111]. CXCL1 is produced by neurons in the brain of Alzheimer’s disease patients [111]. In pathological brain tissue, CXCL1 activates the CXCR2 receptor on neurons, which causes the activation of ERK MAPK and glycogen synthase kinase 3β (GSK3β) leading to the hyperphosphorylation of tau (Figure 3) [111,112]. Then, prolonged exposure of neurons to CXCL1 results in Tau cleavage at Asp^421^ by caspase-3 [112]. Such truncated tau proteins may be the primer of neurofibrillary tangle formation [112]. Hyperphosphorylation also increases the aggregation capacity of tau proteins. CXCR2 activation has been found to increase γ-secretase activity, a protease whose substrate is amyloid precursor protein (APP) [113,114,115]. This causes the release of Aβ which contributes to the formation of amyloid plaques.

CXCL1 may also participate in mechanisms that inhibit the progression of Alzheimer’s disease. Patients with this disease have elevated CXCL1 expression in blood monocytes [116]. In addition, expression of CXCR2 on brain microvascular endothelial cells is increased by Aβ [116]. This indicates that the CXCL1→CXCR2 axis is responsible for transendothelial migration of monocytes into the brain. Such monocytes differentiate into bone marrow-derived microglia, involved in the elimination of Aβ plaque deposition, inhibiting the progression of Alzheimer’s disease [116,117].

### 6.5. Epilepsy

Epilepsy, a brain disease whose major symptoms include unprovoked seizures, is found in about 80 million people in the world [118]. Ligands for CXCR2 contribute to its pathogenesis—as shown, for example, by increased KC expression in the brain of epileptic mice, particularly in the hippocampus [119]. KC decreases astrocytic glutamate reuptake [120], which causes an increase in glutamate concentration at synapses and leads to seizures. CXCL1 does not seem to have the same role in epilepsy in humans as it does in mice. In contrast to TNF-α and CXCL9, CXCL1 levels are not elevated in the cerebrospinal fluid of patients with epilepsy [121]. In comparison, CXCL8 expression is increased in the brain of patients with epilepsy [119] which suggests its role is similar to that of KC in mouse epilepsy. However, further studies in this direction are required.

### 6.6. Herpes Simplex Virus Type 1 (HSV-1) Encephalitis and Herpetic Stromal Keratitis (HSK)

Herpes simplex virus type 1 (HSV-1) is a virus possessing double-stranded DNA of fairly large size: 152 kbp [122]. HSV-1 belongs to the *Alphaherpesvirinae* subfamily, which also includes HSV-2 and Varicella zoster virus (VZV)/HSV-3. It is estimated that approximately half of the population has had contact with HSV-1 [123]. Infection with this virus occurs when fluids containing HSV-1 come into contact with mucous membranes [122]. Then HSV-1 infects the mucosal epithelium and then sensory neurons near the site of primary infection where HSV-1 progresses to latent infections [122,124].

HSV-1 can also enter the brain, where it can cause quite rare cases of severe encephalitis [124] at 0.25–1.2 per 100,000 population per year [124]. HSV-1 encephalitis is associated with an increase in the expression of pro-inflammatory cytokines in the brain [125]. In particular, experiments on mice have shown an increase in KC expression in astrocytes under the influence of HSV-1 [126]. There is also an increase in KC expression in astrocytes and neurons under the influence of pro-inflammatory cytokines such as IL-1α. KC causes infiltration of the brain by neutrophils which results in blood-brain barrier damage and an excessive immune system response, resulting in brain damage and death. The KC→CXCR2 axis and neutrophils do not affect HSV-1 viral load in the brain as shown by experiments in mice [126]. As monocytes and the CCL2→CCR2 axis are responsible for fighting the virus in the brain, the KC→CXCR2 axis represents a convenient therapeutic target against HSV-1 encephalitis in a mouse model. With mouse KC being a paralog for human CXCL1 [5,6,11], it needs to be confirmed if changes in CXCL1 expression occur in patients with HSV-1 encephalitis, to determine its role in this disease in humans.

HSV-1 has also been found to infect the cornea, leading to herpetic stromal keratitis (HSK) [127], often resulting in blindness. Cornea infection by HSV-1 is associated with an increase in KC expression due to the action of IL-17A, as shown by experiments in a mouse model [128]. This leads to infiltration of the cornea by neutrophils [128]—cells involved in fighting the viral infection but also in damaging the cornea [127]. Among other things, neutrophils secrete vascular endothelial growth factor A (VEGF-A) and enhance the action of this growth factor by secreting MMPs, which leads to corneal neovascularization [128]. Depending on the study, either KC [127,128] or MIP-2 [129,130] plays a major role in neutrophil infiltration into the cornea during HSV-1 infection. Both chemokines are CXC chemokines [5,6] and are paralogs to human CXCL1, so the significance of CXCL1 needs to be confirmed in human patients with the aforementioned diseases.

### 6.7. Ischemic Stroke

Ischemic stroke results from brain arterial occlusion followed by reduced blood flow to various parts of the brain [131]. Very often it ends in death or extensive brain damage and disability. It is estimated that nearly 10 million cases of ischemic stroke occur annually, making it one of the most common diseases in the world [131].

CXCL1 expression has been found to be closely associated with ischemic stroke. Therefore, individuals with the T allele of rs3117604, which is located in the CXCL1 promoter, have an increased predisposition to ischemic stroke [132].

Patients with ischemic stroke have elevated levels of CXCL1 in cerebrospinal fluid [133] and at the same time CXCL1 levels are associated with the extent of ischemic stroke. Increased CXCL1 expression has also been shown in brain tissue affected by ischaemic stroke [134]. Nevertheless, depending on the literature cited, CXCL1 levels in the blood in patients with ischemic stroke are either unchanged [133] or lower [135]. With the discussed CXCL1 levels in the blood, this may be related to gender. In women with ischaemic stroke, CXCL1 levels in the blood are lower, while in men they are higher [136].

Expression of KC in the brain at the ischaemic stroke takes place in astrocytes, as shown by experiments on mice (Figure 4) [134]. γδ T cells, which produce IL-17A, are responsible for the induction of KC expression. Expression of KC in astrocytes also depends on TNF-α produced by macrophages [134]. At the same time, CD4^+^ T cells produce interferon-γ (IFN-γ), which induces TNF-α expression in macrophages. 

Expression of ligands for CXCR2 during ischemic stroke may also depend on miRNAs. In rats, an ischemic stroke is associated with promoter hypermethylation of miR-532-5p [137], which reduces the expression of the miRNA that is directly downregulating the expression of the rat paralog for human CXCL1. That means that this downregulation of miR-532-5p expression induces an upregulation of CXCL1 expression. In contrast, during ischemic stroke in mice, there is a downregulation of miR-429 expression in brain microvascular endothelial cells [138]. This miRNA downregulates the expression of KC, the mouse paralog for human CXCL1, which results in increased expression of KC. As a consequence of increased CXCR2 ligand expression, neutrophils infiltrate into brain tissue undergoing ischemic stroke. It is not known whether these cells have a destructive effect on brain tissues in patients with ischemic stroke. Studies in mice have shown that a CXCR2 receptor inhibitor does not affect the level of motor dysfunction after ischemic stroke [139]. However, neutrophils can disrupt the blood-brain barrier (BBB) integrity through elastase secretion [140]. Because the contribution of neutrophils to brain damage after ischemic stroke has not been sufficiently investigated, more thorough studies in this direction are needed.

### 6.8. Major Depression

Major depression is a mental illness that affects approximately 6% of the population and its pathogenesis may be related to inflammation [141]. In mice, chronic stress causes activation of the inflammasome in the hippocampus [142,143], increasing the production of the pro-inflammatory cytokines IL-1β and interleukin-18 (IL-18). This leads to inflammatory responses associated with increased KC production [143]. KC, through activation of its receptor CXCR2 and subsequent activation of GSK3β, induces depression-like behaviors.

The mechanism of CXCL1 involvement in major depression in humans appears to differ significantly from the animal model. Depressed suicidal persons experience a reduction in CXCL1 expression in the prefrontal cortex, similar to the reduction in the expression of CXCL2 and CXCL3 [144]. Plasma CXCL1 levels are also reduced in depressed humans [145], particularly in elderly patients [146] and adolescents [147]. One paper shows that in elderly patients, plasma CXCL1 may be slightly increased in depression, although the results were not statistically significant [148]. In contrast, depressed patients have been shown to have much higher levels of CXCL7 and CXCL8 in their blood [149]. It is likely that these chemokines have a function in depression that KC plays in mice, but another mechanism cannot be ruled out.

There are no studies showing correlations between depression on CXCL1 expression in the hippocampus in humans. It is possible that, as in experimental animals, depression is associated with an increase in CXCL1 expression in this brain structure in humans. If, in contrast, the human hippocampus showed a decrease in CXCL1 expression, it would indicate that CXCL1 plays an important role in brain function in humans and a decrease in the expression of this chemokine could cause depression. This would likely be related to hippocampal neurogenesis [104] or impaired oligodendrocyte function and myelination [39,150,151].

### 6.9. Multiple Sclerosis

Multiple sclerosis is a neurodegenerative and autoimmune disease of the brain and spinal cord, with an estimated incidence of 0.2 to 240 people per 100,000, depending on the population studied [152]. Myelin-reactive T cells are the main element responsible for the pathogenesis of this disease [153]. Following the action of these cells, demyelination and inflammatory reactions result in the dysfunction of the nervous tissue.

CXCL1 is a significant element in the course of multiple sclerosis. This chemokine has been shown to have both neuroprotective and inhibitory properties in the progression of multiple sclerosis [39], as well as being one of the elements contributing to neurodegeneration [154]. Nevertheless, it seems that the destructive properties of CXCL1 are predominant, as in patients with multiple sclerosis, plasma CXCL1 levels are correlated with clinical disability [155].

CXCL1 expression is increased in the brains of patients with multiple sclerosis [150]. In particular, it is found in areas of demyelination [154,156]. Elevated CXCL1 levels have also been shown in multiple sclerosis patients in blood [155] and cerebrospinal fluid [157,158]. However, these data are debatable as some studies have shown that CXCL1 levels in cerebrospinal fluid were not different in patients with multiple sclerosis [159,160]. This discrepancy may be due to the fact that CXCL1 could be a significant factor in the development of multiple sclerosis in just the early stages of the disease [161,162].

In multiple sclerosis, CXCL1 is produced by activated microglia [163] and astrocytes (Figure 5) [150,151]. Also, CXCL1 expression is dependent on CD4^+^ T helper type 17 (Th17) cells which produce IL-17 [155,164,165]. This cytokine increases the expression of CXCL1. IL-17 has toxic effects on oligodendrocytes [166]. IL-17 appears to be significant in the early stages of multiple sclerosis [161]. As the disease progresses and treatment is given, the concentration of IL-17 in cerebrospinal fluid decreases. There are also indications that CXCL1 expression and disease progression in patients with multiple sclerosis may also be inhibited by IFN-γ [167]. In mice, IFN-γ inhibits the expression of MIP-2 but not KC [165]. However, as it is very difficult to find the exact counterparts of these two CXC chemokines [5], there is a need for experiments investigating the effect of IFN-γ on CXCL1 expression in human neural tissue during multiple sclerosis. IFN-γ increases the production of nitric oxide (NO) which enhances the ability of neutrophils to inhibit autoreactive T cells [167,168], which inhibits the progression of multiple sclerosis.

Inflammation has a destructive effect on oligodendrocyte progenitor cells as shown by experiments on human embryonic stem cell-derived oligodendrocyte progenitor cells [169] and murine oligodendrocyte progenitor cells [170]. This is related to the induction of apoptosis of these cells by CXCL10/γ interferon inducible protein 10 (IP-10), a chemokine whose expression is induced by IFN-γ [169,170]. In contrast, CXCL1 and thus CXCR2 receptor activation inhibits CXCL10/IP-10-dependent apoptosis of oligodendrocyte progenitor cells [169,170]. IFN-γ and CXCL10/IP-10 are elevated in the cerebrospinal fluid of patients with multiple sclerosis [158]. Therefore, the adverse effect of CXCL10/IP-10 on oligodendrocyte progenitor cells appears to occur in patients with multiple sclerosis. Elevated CXCL1 expression occurs in areas of demyelination [154,156]. Research on oligodendrocyte precursors has shown that CXCL1 induces proliferation of these cells [38,39]. At the same time, this chemokine inhibits oligodendrocyte precursor cell migration [39,103]. As a consequence, oligodendrocyte precursor cells accumulate in areas of demyelination. Then, these cells participate in remyelination [39,150,151]. Finally, CXCL1 can inhibit multiple sclerosis by causing neutrophil infiltration into the brain. These cells are able to inhibit the activity of autoreactive T cells [168], which inhibits the progression of multiple sclerosis.

CXCL1 also participates in the progression of multiple sclerosis [154]. CXCR2 activation inhibits oligodendrocyte precursor cell differentiation, and so ligands of this receptor inhibit remyelination in this way [154]. CXCL1 also causes neutrophil infiltration into the central nervous system [162,164,165,171]. Neutrophils cause neurodegeneration, as they produce ROS that acts destructively on the neural tissue [172,173]. This effect is enhanced by CXCR2 activation [173]. These cells also produce proinflammatory cytokines and thus increase neuroinflammation in patients with multiple sclerosis. The cells described also cause a breakdown of the BBB [172]. Due to the importance of neutrophils in the course of multiple sclerosis, an elevated neutrophil-lymphocyte ratio (NLR) is associated with faster progression and a greater disability in patients with this disease [174,175], with neutrophils appearing in the cerebrospinal fluid of multiple sclerosis patients in the early stages of the disease [161]. Subsequently, the level of these cells decreases with treatment and length of the disease, which shows that neutrophils are an important pathogenic factor only in the initial stages of multiple sclerosis [161,162].

### 6.10. Neuromyelitis Optica

Neuromyelitis optica is a neurodegenerative and autoimmune disease. Auto-antibodies to aquaporin 4 (AQP4) are responsible in over 80% of cases of this disease [176]. The incidence of this disease is estimated to be between 0.5 and 10 cases per 100,000 population, depending on the country. These auto-antibodies directly cause an increase in CXCL1 secretion from astrocytes [177]. Therefore, CXCL1 levels are elevated in neuromyelitis optica in cerebrospinal fluid [160] and in serum [177]. It was found that CXCL1 levels are not correlated with patient clinical severity [160], which shows that CXCL1 can only be a marker for patients with neuromyelitis optica, but does not affect the development and severity of the disease. Also, studies in animal models have shown that CXCR2, the receptor for the chemokine in question, is irrelevant in the development of neuromyelitis optica [178]. However, studies in an animal model of neuromyelitis optica have shown that neutrophil proteases are an important factor in the pathophysiology of this disease [179]. CXCL1 is a chemokine that induces neutrophil recruitment. Therefore, it may play an indirect role in the development of neuromyelitis optica by recruiting these cells into the neural tissue.

### 6.11. Neuropathic Pain and Sickness Behaviors

Chemokines, such as CXCR2 ligands, are important in the development of neuropathic pain. However, due to bioethical constraints, all knowledge about the importance of CXC chemokines in the development of neuropathic pain is based on experiments on laboratory animals. Due to the fact that CXCR2 ligand systems differ significantly between humans and mice, the presented mechanism of neuropathic pain gives only a hint of the role that CXCL1 may play in this disease in humans.

Following nerve injury in mice, KC expression increases in spinal astrocytes [180,181,182]. This process is dependent on TNF-α which also increases the production of IL-1β. This pro-inflammatory cytokine increases cyclooxygenase-2 (COX-2) activity and the production of prostanoids which are also responsible for the sensation of pain [183]. KC also induces inflammatory responses, in particular the production of TNF-α, IL-1β, IL-6 and CCL2 [182,184]. Nerve injury is also associated with an increase in CXCR2 expression in dorsal horn neurons [180]. Similarly, following traumatic brain injury, increased CXCR2 expression has been observed in spinal cord neurons [185].

CXCR2 activation on neurons and spinal microglia has been found to lead to neuropathic pain [180,181,186,187]. Specifically, KC results in the release of sympathetic amines [183], increased signaling via the transient receptor potential vanilloid type 1 (TRPV1) channel on sensory neurons [188], upregulation of Na^+^ currents [189] and upregulation of K^+^ currents in sensory neurons [190]. Another mechanism by which CXCR2 ligands cause neuropathic pain is through increased release of calcitonin gene-related peptide (CGRP), as confirmed by studies on chemokine CINC-1 in rats [191]. CGRP is thus responsible for pain generation. Also, CXCR2-dependent KC increases N-methyl-D-aspartate (NMDA)-induced currents in lamina II neurons in a mouse model [181].

It has been found that KC can also cause neuropathic pain by recruiting neutrophils [182,192,193]. These cells secrete cathepsin E which is involved in pain [193]. However, activation of CXCR2 on neutrophils results in the release of endogenous opioids from these cells that reduce pain [194]. KC is also important in the formation of post-surgical pain, as infiltration of neutrophils near surgical wounds, cells that cause post-surgical pain, is dependent on the KC→CXCR2 axis in mice [195].

Also, studies with laboratory animals show that CXCR2 ligands reduce spontaneous activity [196]. This indicates that CXCL1 may be associated with poor mental status and sickness behaviors of patients with multiple sclerosis, as well as other neuroinflammatory diseases.

### 6.12. Prion Diseases

Prion diseases form a group of neurodegenerative diseases caused by scrapie-associated prion protein (PrP^Sc^) [197]. This protein has a pathogenic misfolding that replicates by converting the structure of cellular prion protein (PrP^C^) into PrP^Sc^. This is followed by aggregation of PrP^Sc^ in neural tissue which leads to spongiform degeneration. Examples of prion diseases include sporadic Creutzfeldt-Jakob disease (sCJD), fatal familial insomnia and kuru. An important element in the course of prion diseases is neuroinflammation [198] involving microglia and astroglia [199,200,201]. Studies in mice have shown that both cell types produce p40 subunit of IL-12 (IL-12p40) and CXCL10/IP-10 [199]. In contrast, astroglia produce cytokines such as IL-1β, IL-6, IL-12p70, CCL2, CCL3, CCL5 and KC.

As KC is a murine paralog for human CXCL1 [5,11], it may be suspected that humans with prion diseases have an increase in CXCL1 expression in the brain, although that needs to be confirmed by further research. We also need more research on the significance of CXCL1 in the course of spongiform degeneration of the brain.

CXCL1 does not play an important role in prion diseases. In multiple sclerosis, an increase in CXCL1 expression in the brain has a destructive effect on neural tissue through infiltration of neural tissue by neutrophils [155,173]. Even when infiltration of neutrophils occurs in prion diseases, PrP^Sc^ reduces the activity of these cells [202] and therefore, these cells do not have a destructive effect on neural tissue and at the same time have no function in the course of prion diseases. Thus, the main property of CXCL1, which is the effect on neutrophils, does not play an important role in the course of prion diseases.

### 6.13. Tick-Borne Encephalitis (TBE) and Ticks

Tick-borne encephalitis (TBE) is a disease caused by the tick-borne encephalitis virus (TBEV) [203]. Its genetic material is a positive-sense single-stranded RNA of approximately 11kb in length. This virus belongs to the genus *Flavivirus*. It infects animals in central and eastern Europe and northern Asia. In humans, TBEV infection usually gives mild symptoms; however, some 2% of cases of TBEV infection are fatal. TBEV is a neurotropic pathogen that attacks neural tissue, including the brain. The vector for this virus is the *Ixodes* sp. tick, which means the virus enters the body of the host during feeding by these parasites. 

At the same time, the feeding of the tick involves a strong immune response [204]. The tick’s bite initially induces an increased expression of pro-inflammatory cytokines that are chemoattractants for lymphocytes, macrophages and neutrophils (Figure 6). However, the expression of proinflammatory cytokines is downregulated [204] and chemokines are inactivated by tick salivary compounds [205]. Among these compounds are evasins [206,207,208,209], glycoproteins commonly found in the saliva of various tick species [207]. In particular, evasin-3 specifically binds to CXC chemokines, including CXCL1 [206]. CXCL1 in such a complex cannot activate its CXCR2 receptor but can still be bound to glycosaminoglycans [209]. This results in the inactivation of CXCL1 and reduced infiltration of the tick feeding site by neutrophils [206,208].

Other evasins inactivate other groups of chemokines. For example, Evasin-1 inactivates the proinflammatory chemokines CCL3 and CCL4 [206]. Evasins and other tick salivary compounds are able to inhibit the immune response at the tick feeding site [205,208]. For this reason, the tick can feed unnoticed by its host for a prolonged time.

TBEV is a neurotropic pathogen causing a severe neuropathological disorder [203]. One component of this disease is neuroinflammation [210], associated with elevated levels of CXCL1 and CXCL8/IL-8 in cerebrospinal fluid [211]. The levels of these chemokines are correlated with neutrophil infiltration of neural tissue [211]. There is also an increase in IL-17 levels [211], a cytokine that increases CXCL1 expression [32,33]. This indicates that TBEV infection of neural tissue cells induces an increase in IL-17 production, which then causes an increase in CXCL1 recruiting neutrophils to the sites of the TBEV infection to control the disease.

### 6.14. Traumatic Spinal Cord Injury

Traumatic spinal cord injury involves damage to the spinal cord as a result of a violent accident or fall. It is estimated that there are approximately 17,000 cases of traumatic spinal cord injury per year in the United States alone [212]. Approximately one week after a traumatic spinal cord injury, patients experience an increase in blood CXCL1 levels [213]. This is the result of an increase in the expression of IL-1β in the liver, a pro-inflammatory cytokine which increases the expression of CXCL1 in this organ [214], triggering the mobilization of neutrophils from the bone marrow [62]. A similar mechanism occurs at the site of the spinal cord injury, where an increase in IL-1β production elevates CXCL1 expression [215,216]. This increased expression of CXCL1 in the spinal cord is not TNF-α dependent [215]. These processes are followed by infiltration of the spinal cord by neutrophils [214]. These cells secrete elastase, which causes additional damage to the spinal cord [217]. This leads to various secondary dysfunctions in patients with traumatic spinal cord injury.

### 6.15. West Nile Fever

West Nile fever is caused by West Nile Virus (WNV) from the genus *Flavivirus* [218]. Its genetic material is positive-sense single-stranded RNA with a length of about 11 kb. WNV is not a dangerous pathogen for humans with a healthy immune system where it causes only mild symptoms [218].

WNV acts as a neurotropic pathogen that affects the nervous system and also other tissues and organs such as the skin, kidney and gastrointestinal tract. WNV is a zoonotic pathogen that is transmitted to humans by mosquitoes, mainly by *Culex* sp. For this reason it is found in warm countries in Africa, Asia, Australia and Europe. When bitten by the mosquito, the virus is introduced into the skin [219]. It infects Langerhans cells and keratinocytes, where it replicates [218], causing inflammatory responses such as increased expression of pro-inflammatory cytokines and chemokines recruiting neutrophils: CXCL1, CXCL2 and CXCL8/IL-8 [219]. This reaction of the immune system results in the eradication of WNV infection.

Anti-inflammatory substances in the mosquito saliva reduce inflammatory responses [219], which allows WNV replication in the skin in the first stage of infection. If the infection is not suppressed by the immune system, WNV then enters the lymph nodes where it further replicates [218]. WNV causes increased expression of chemokines that attract neutrophils, including CXCL1 [220]. This is followed by the recruitment of neutrophils to the sites of WNV infection where these cells are also infected [220]. The virus replicates in neutrophils which then enter the blood and constitute a reservoir and a type of vector that causes dissemination of WNV throughout the body. However, at the late stages of infection, non-infected neutrophils are an important part of the fight against WNV infection [220].

WNV is a neurotropic pathogen that can get into the brain and spinal cord, although this form of infection is rare in humans [218]. There are many theories, some of which are mutually exclusive, about the mechanism by which WNV enters the neural tissue [218]. One of them suggests that WNV-infected neutrophils migrate into the neural tissue in a kind of ‘Trojan horse’ mechanism [221]. In the neural tissue, WNV infects neurons and astrocytes but not microglia cells [222,223], resulting in the death of infected cells. Then microglial cells phagocytose infected cells and cell debris [224] which causes the activation of the microglia and thus secretion of many pro-inflammatory cytokines and chemokines, including CXCL1 [224]. This leads to the infiltration of infected neural tissue by immune cells including neutrophils [225], resulting in the eradication of WNV infection.

## 7. Directions of Further Research

The role of CXCL1 in physiology and in non-cancer diseases has been fairly well studied. Nevertheless, the biggest shortcoming in knowledge of this chemokine is the huge difference in CXCR2 ligand systems between humans and rodents. For this reason, human CXCL1 does not have functions identical to those of murine KC and rat CINC-1. The results obtained in studies on animals cannot be completely and unqualifiedly compared to a human patient’s condition during a given disease. For this reason, until new research models are invented and disseminated, certain areas of knowledge about CXCL1 are still in the realm of conjecture.

One direction of CXCL1 research should be the application of existing knowledge in practice. CXCL1 plays an important role in the pathogenesis of many non-cancer diseases. Therefore, it is possible to use either CXCR2 inhibitors or anti-CXCL1 antibodies in the therapy of these diseases. As there are 7 chemokines that activate CXCR2, it seems that the use of CXCR2 inhibitors is a better option. However, the chemokines that are CXCR2 activators differ in their sites of expression, among other things. For this reason, antibodies to individual CXCR2 activators may also be a convenient therapeutic option, provided we learn the exact differences between individual CXCR2 activators in a given disease.

## Figures and Tables

**Figure 1 ijms-23-04205-f001:**
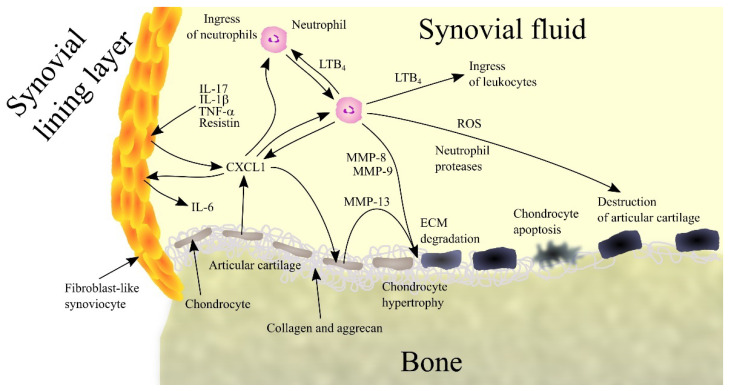
Importance of CXC motif chemokine ligand 1 (CXCL1) and neutrophils in rheumatoid arthritis. Patients with rheumatoid arthritis have elevated levels of interleukin-1β (IL-1β), tumor necrosis factor α (TNF-α), interleukin-17 (IL-17) and resistin in synovial fluid. These factors induce CXCL1 expression in fibroblast-like synoviocytes (FLS), chondrocytes and neutrophils, which leads to an increase in CXCL1 levels in synovial fluid. CXCL1 causes an increase in interleukin-6 (IL-6) expression in FLS. This chemokine causes leukotriene B_4_ (LTB_4_) synthesis in neutrophils and thus an ingress of more neutrophils and leukocytes into the joints. CXCL1 also causes an increase in matrix metalloproteinase-13 (MMP-13) expression in chondrocytes, which leads to extracellular matrix (ECM) articular cartilage degradation. Also responsible for this process are matrix metalloproteinase-8 (MMP-8) and matrix metalloproteinase-9 (MMP-9) produced by neutrophils. CXCL1 also causes chondrocyte hypertrophy and apoptosis. Finally, neutrophils secrete reactive oxygen species (ROS) and neutrophil proteases into synovial fluid. All these processes and factors lead to the destruction of articular cartilage and symptoms of rheumatoid arthritis. Abbreviations: CXCL1—CXC motif chemokine ligand 1; ECM—extracellular matrix; IL-1β—interleukin-1β; IL-6—interleukin-6; IL-17—interleukin-17; LTB_4_—leukotriene B_4_; MMP—matrix metalloproteinase; ROS - reactive oxygen species; TNF-α - tumor necrosis factor α; Source: own elaboration.

**Figure 2 ijms-23-04205-f002:**
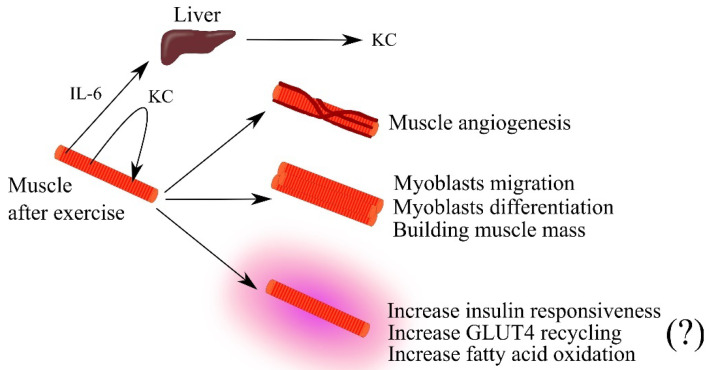
The importance of keratinocyte-derived chemokine (KC) in skeletal muscle physiology. Intense effort induces an increase in the expression of interleukin-6 (IL-6) and KC in muscle. IL-6 travels through the bloodstream to the liver where it increases KC expression. This leads to an increase in blood levels of this chemokine. KC secreted by muscle acts in an autocrine manner, causing muscle angiogenesis, the growth of muscle mass by acting on myoblasts, and increasing muscle efficiency by increasing insulin responsiveness and fatty acid oxidation in the muscle. Abbreviations: (?)—mechanism in question; GLUT4—glucose transporter 4; IL-6—interleukin-6; KC—keratinocyte-derived chemokine; Source: own elaboration.

**Figure 3 ijms-23-04205-f003:**
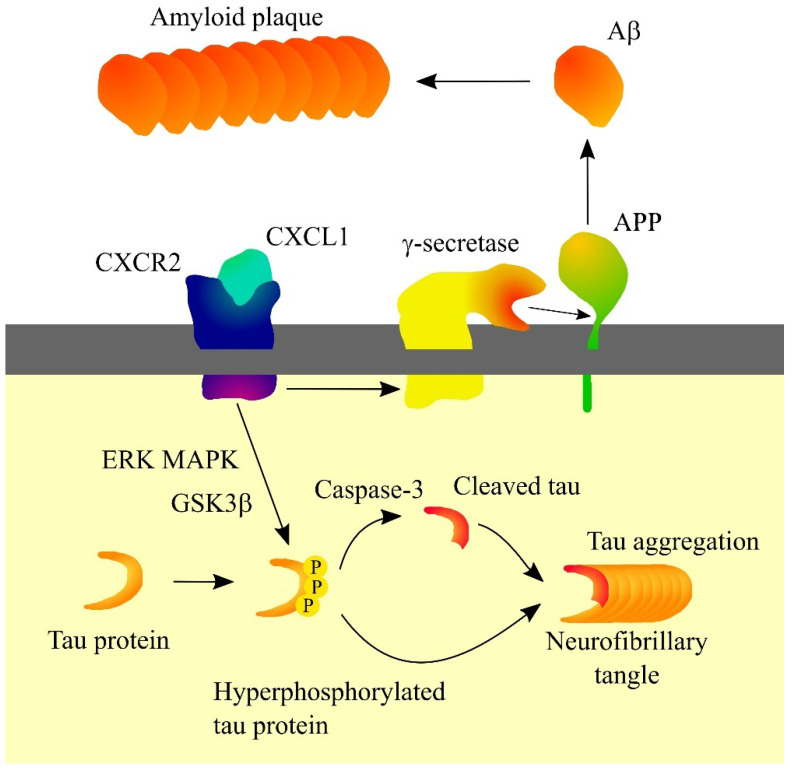
Significance of CXC motif chemokine ligand 1 (CXCL1) in Alzheimer’s disease. CXCL1, through its receptor CXC motif chemokine receptor 2 (CXCR2), causes hyperphosphorylation of tau protein which leads to a proteolytic cleavage of tau protein, which enables the formation of a neurofibrillary tangle. CXCR2 also increases γ-secretase activity. This results in increased release of amyloid β (Aβ) and increased amyloid plaque formation. Abbreviations: Aβ—amyloid β; APP—amyloid precursor protein; CXCL1—CXC motif chemokine ligand 1; CXCR2—CXC motif chemokine receptor 2; ERK—extracellular signal-regulated kinase; GSK3β—glycogen synthase kinase 3β; MAPK—mitogen-activated protein kinase; PPP—hyperphosphorylation; Source: own elaboration.

**Figure 4 ijms-23-04205-f004:**
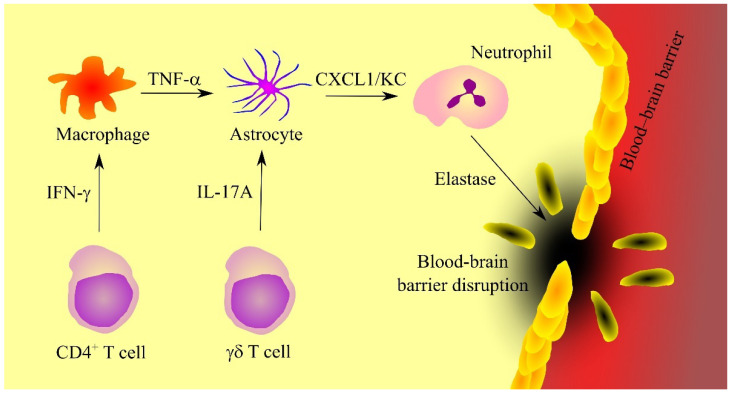
The significance of CXC motif chemokine ligand 1 (CXCL1)/keratinocyte-derived chemokine (KC) in ischemic stroke. Ischaemic stroke is associated with neuroinflammation caused by γδ T cells and CD4^+^ T cells that respectively secrete interleukin-17A (IL-17A) and interferon-γ (INF-γ). INF-γ increases tumor necrosis factor α (TNF-α) production in macrophages. TNF-α and IL-17A increase the secretion of CXCL1/KC in astrocytes. This chemokine causes recruitment of neutrophils, cells which secrete elastase, an enzyme that causes blood-brain barrier disruption. Abbreviations: CXCL1—CXC motif chemokine ligand 1; IL-17A—interleukin-17A; INF-γ—interferon-γ; KC—keratinocyte-derived chemokine; TNF-α—tumor necrosis factor α; Source: own elaboration.

**Figure 5 ijms-23-04205-f005:**
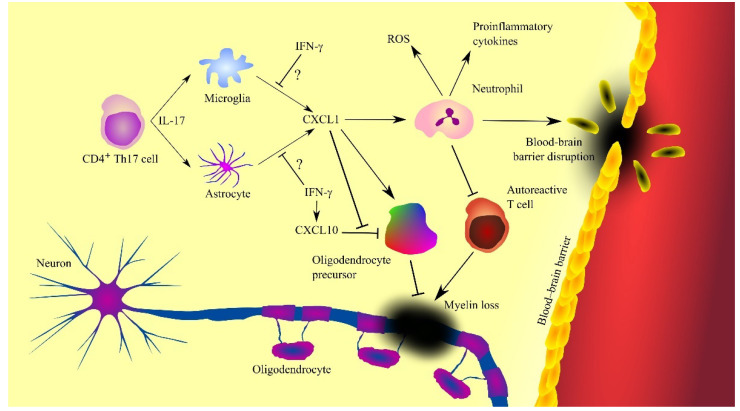
Involvement of CXC motif chemokine ligand 1 (CXCL1) in the mechanisms of multiple sclerosis. CXCL1 expression is increased by interleukin-17 (IL-17) produced by T helper type 17 (Th17) cells, which triggers the recruitment of neutrophils, cells contributing to the progression of the disease by producing reactive oxygen species (ROS), proinflammatory cytokines and causing blood-brain barrier (BBB) disruption. However, neutrophils can also inhibit the destructive effects of autoreactive T cells. CXCL1 also acts on oligodendrocyte precursor cells by inducing their proliferation and inhibiting CXC motif chemokine ligand 10 (CXCL10)-induced apoptosis of these cells. Oligodendrocyte precursor cells differentiate into oligodendrocytes, leading to remyelination and disease regression. Abbreviations: CXCL1—CXC motif chemokine ligand 1; CXCL10—CXC motif chemokine ligand 10; IFN-γ—interferon-γ; IL-17—interleukin-17; ROS—reactive oxygen species; Th17—T helper type 17 cells; Source: own elaboration.

**Figure 6 ijms-23-04205-f006:**
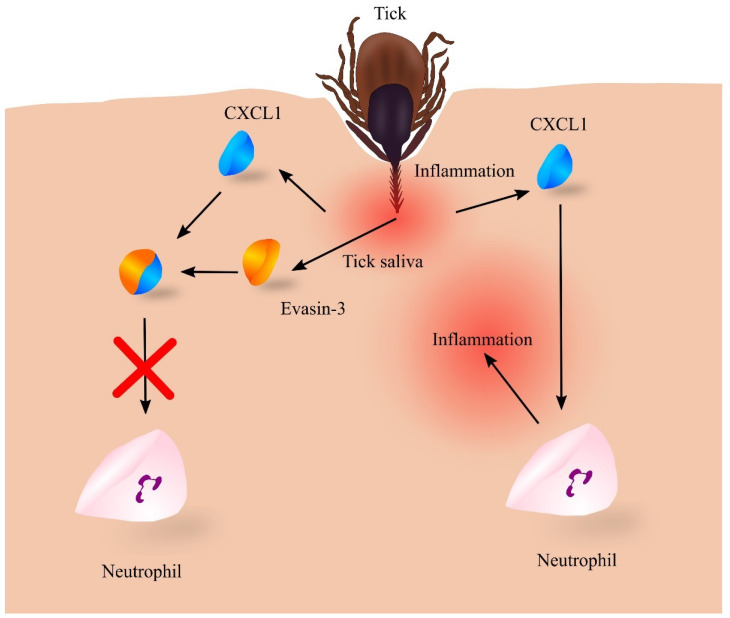
Role of evasin-3 in inhibiting the inflammatory response to tick feeding. Tick feeding leads to inflammatory responses, which induce an increase in the expression of chemokines including CXC motif chemokine ligand 1 (CXCL1), which recruits neutrophils to the vicinity of the feeding tick. This leads to even greater inflammatory reactions, itching of the skin, and removal of the tick by the host. However, the tick introduces its saliva into the site of the puncture while feeding, which contains a number of proteins, including evasin-3, which binds to CXCL1, resulting in the loss of its biological properties. No neutrophil recruitment then occurs and the tick can feed unnoticed for some time. Source: own elaboration.

## Data Availability

Not applicable.

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
