# Peer review of "The Importance of CXCL1 in Physiology and Noncancerous Diseases of Bone, Bone Marrow, Muscle and the Nervous System"

_ijms, 2022, doi:10.3390/ijms23084205_

Round 1

Reviewer 1 Report

Dear Authors,

Topical and large review, also very logic and commonly nicely organized. Absolutely amazing References part (didnt see such one for very long time, this part enjoyed me very much)! However, I have some minor comments regarding some places...

1) I would like to ask for slight changes in the section 2. Please, develop clear plan here of the done work, - with some subsections/separation of paragraphs  with clarification of data bases used for the literature search, time when it was done (from-to), key words, sources inclusion/exclusion criteria, human/animal studies... You can leave some commentaries, but in way, how the section is developed now, its a little bit intermixed, confusing and problematically comprehensible...

(Perhaps it might be even “Method” section, describing how the comprehensive search of publications has been performed (eg: database used, syntax applied, period of the search …etc.). I would also invite the authors to describe the number of citations found, what all can prepare the reader for the next really nicely described following issues.)

2) Please, decipher the abbreviations after the each Fig for all 6 Figs what you have. Please, also indicate the source from what the Figs were adopted after the each of them;

3) Finally, you have 24 previous century sources (out of 225) in the References part. Well, not very many, but I would like to ask or to re-place them with more topical, or to remove, or to add then the sentence about the historical aspect in every place, where you give the "old" reference. They simply do not fit in this really nice review and also are not acceptable for high IF scientific Journal.

Author Response

Dear Authors,

Topical and large review, also very logic and commonly nicely organized. Absolutely amazing References part (didnt see such one for very long time, this part enjoyed me very much)! However, I have some minor comments regarding some places...

1) I would like to ask for slight changes in the section 2. Please, develop clear plan here of the done work, - with some subsections/separation of paragraphs  with clarification of data bases used for the literature search, time when it was done (from-to), key words, sources inclusion/exclusion criteria, human/animal studies... You can leave some commentaries, but in way, how the section is developed now, its a little bit intermixed, confusing and problematically comprehensible...

(Perhaps it might be even “Method” section, describing how the comprehensive search of publications has been performed (eg: database used, syntax applied, period of the search …etc.). I would also invite the authors to describe the number of citations found, what all can prepare the reader for the next really nicely described following issues.)

We have added a methodology used to search for publications that were needed to write this review.

2) Please, decipher the abbreviations after the each Fig for all 6 Figs what you have. Please, also indicate the source from what the Figs were adopted after the each of them;

Abbreviations in figure descriptions have been added along with the sources of the figures.

3) Finally, you have 24 previous century sources (out of 225) in the References part. Well, not very many, but I would like to ask or to re-place them with more topical, or to remove, or to add then the sentence about the historical aspect in every place, where you give the "old" reference. They simply do not fit in this really nice review and also are not acceptable for high IF scientific Journal.

Regarding citation of pre-2000 articles, we performed an update of 9 publications from the bibliography to more recent publications as recommended by the reviewer. However, some of the publications identified by the reviewer are still cited because:

- they were the first to describe the process in question. We then cited newer papers. In other words, in one place we cited 2-3 papers where the first one was from before 2000 and the following were the most recent papers that described the exact disease/physiological mechanism (5 papers)

- they were the first to identify/characterize CXCL1, KC or CINC-1 (4 papers)

- precise studies on the activation of CXCR1 and CXCR2 receptors by CXCR2 ligands were conducted before 2000. Subsequently, these publications were cited with varying accuracy, e.g., some recent articles incorrectly indicate that CXCL1 is a ligand exclusively for CXCR2, even though CXCL1 is also a ligand for CXCR1 (at high concentrations) and ACKR1. For this reason, we thought it best to cite articles that calculated the exact parameters in which CXCL1 binds to and activates CXCR1 and CXCR2. (4 papers)

- some studies (2 papers) describe the consequences of CXCL1 action, which have not been studied again. This means that the results of those papers have never been refuted. We believe that omitting these papers may limit the understanding of CXCL1 action in some diseases.

Reviewer 2 Report

In this review, Korbecki et al. have discussed the importance of CXCL1 (CXC motif chemokine ligand 1) in physiology of bone, bone marrow, muscle and the nervous system and their respective disease conditions. The authors have addressed one of the important issue around CXCL1  that human CXCL1 does not have same functions identical to those of murine and rat such as KC and CINC-1, respectively. Due to that reason, studies obtained in animals can not be completely compared to a human patients' conditions. In this review, the authors have provided details in aforementioned systems across different species models and summerised the findings. For this reason, the authors argue that new research models should be invented.

The review is well written and covered very wide spectrum. The models are shown when it is necessary to better understand the working mechanisms. For this reason, the review can be accepted in its current form.

Author Response

We are very grateful for review of our manuscript.